Clinical dosage of lidocaine does not impact the biomedical outcome of sepsis-induced acute respiratory distress syndrome in a porcine model

Rissel René Rene.Rissel@unimedizin-mainz.de
Moellmann Christian
Albertsmeier Victoria
Renz Miriam
Ruemmler Robert
Kamuf Jens
Hartmann Erik K.
Ziebart Alexander
Department of Anaesthesiology, Medical Centre of the Johannes Gutenberg-University , Mainz , Germany
Tharmalingam Nagendran
Electronic publication date: 2023 Aug 21
Publication date: 2023
Volume: 11
Electronic Location ID: e15875
Received 2023 Mar 26; Accepted 2023 Jul 18
Copyright: © 2023 Rissel et al.
Copyright year: 2023
Copyright holder: Rissel et al.
License: This is an open access article distributed under the terms of the Creative Commons Attribution License, which permits unrestricted use, distribution, reproduction and adaptation in any medium and for any purpose provided that it is properly attributed. For attribution, the original author(s), title, publication source (PeerJ) and either DOI or URL of the article must be cited.
License URL: https://creativecommons.org/licenses/by/4.0/

Keywords: ARDS, Lidocaine, Inflammation, Animal model, Sepsis

Funding: The authors received no funding for this work.

==============================
Background

Sepsis is a common disease in intensive care units worldwide, which is associated with high morbidity and mortality. This process is often associated with multiple organ failure including acute lung injury. Although massive research efforts have been made for decades, there is no specific therapy for sepsis to date. Early and best treatment is crucial. Lidocaine is a common local anesthetic and used worldwide. It blocks the fast voltage-gated sodium (Na+) channels in the neuronal cell membrane responsible for signal propagation. Recent studies show that lidocaine administered intravenously improves pulmonary function and protects pulmonary tissue in pigs under hemorrhagic shock, sepsis and under pulmonary surgery. The aim of this study is to show that lidocaine inhalative induces equivalent effects as lidocaine intravenously in pigs in a lipopolysaccharide (LPS)-induced sepsis with acute lung injury.

Methods

After approval of the local State and Institutional Animal Care Committee, to induce the septic inflammatory response a continuous infusion of lipopolysaccharide (LPS) was administered to the pigs in deep anesthesia. Following induction and stabilisation of sepsis, the study medication was randomly assigned to one of three groups: (1) lidocaine intravenously, (2) lidocaine per inhalation and (3) sham group. All animals were monitored for 8 h using advanced and extended cardiorespiratory monitoring. Postmortem assessment included pulmonary mRNA expression of mediators of early inflammatory response (IL-6 & TNF-alpha), wet-to-dry ratio and lung histology.

Results

Acute respiratory distress syndrome (ARDS) was successfully induced after sepsis-induction with LPS in all three groups measured by a significant decrease in the PaO2/FiO2 ratio. Further, septic hemodynamic alterations were seen in all three groups. Leucocytes and platelets dropped statistically over time due to septic alterations in all groups. The wet-to-dry ratio and the lung histology showed no differences between the groups. Additionally, the pulmonary mRNA expression of the inflammatory mediators IL-6 and TNF-alpha showed no significant changes between the groups. The proposed anti-inflammatory and lung protective effects of lidocaine in sepsis-induced acute lung injury could not be proven in this study.

Introduction

Sepsis is a life-threatening condition characterized by a dysregulated host response to infection, resulting in organ dysfunction (Singer et al., 2016). It is a common and severe disease in critical care medicine worldwide and is associated with high morbidity and mortality rates, often leading to multiple organ failure, including acute lung injury (Alberti et al., 2002). Despite decades of research, there is currently no specific therapy for sepsis, underscoring the need for early and optimal treatment. The latest guidelines from the Surviving Sepsis Campaign emphasize the individual nature of sepsis and highlight various general therapy concepts (Evans et al., 2021).

Given the urgent need for effective treatments for sepsis, various medical concepts are being researched to identify breakthrough therapies. One promising approach is the use of lidocaine. Lidocaine is a commonly used local anesthetic that blocks fast voltage-gated sodium (Na+) channels in neuronal cell membranes, which are responsible for signal propagation. Lidocaine also acts as an anti-arrhythmic agent by blocking the voltage-gated Na+ channels in the heart muscle (Beaussier et al., 2018). However, lidocaine may cause side effects such as sleepiness, confusion, cardiac arrhythmia, and vomiting (Caracas et al., 2009). Recent studies have shown that intravenous administration of lidocaine can improve pulmonary function and protect pulmonary tissue in pigs with hemorrhagic shock, sepsis, or undergoing pulmonary surgery (Romera et al., 2021; Kodama et al., 2000; Berger et al., 2014). Lidocaine has also been shown to attenuate acute respiratory distress syndrome (ARDS) and increase anti-inflammatory effects Although the anti-inflammatory effects of lidocaine have been observed with intravenous administration, recent studies have suggested that nebulized lidocaine may also have the potential to prevent airway inflammation (Serra et al., 2012). However, scientific studies on this effect remain limited, and a direct comparison of the effectiveness of both administration methods in the context of sepsis has yet to be conducted.

The objective of this study is to investigate whether the inhalation of lidocaine can produce effects comparable to those achieved with intravenous administration in a pig model of lipopolysaccharide (LPS)-induced sepsis and acute lung injury. This study aims to fill the existing knowledge gap and contribute to the development of more effective therapies for sepsis.

Materials and Methods

The protocol used in this study was approved by the State and Institutional Animal Care Committee (Rhineland-Palatinate, Koblenz, Germany; ID G16-1-015) in accordance with the ARRIVE guidelines (Kilkenny et al., 2010). The research involved 32 pigs (30 ± 2.5 kg) acquired from a local farmer and transported to the laboratory under sedation with azaperone and midazolam administered intramuscularly.

Anesthesia & instrumentation

Anesthesia was induced and maintained through continuous infusion of propofol and fentanyl, with atracurium used intravenously solely to facilitate endotracheal intubation. Basic monitoring, including pulse oximetry (Radical 7; Masimo, Irvine, CA, USA) and invasive blood pressure (S/5, GE-Datex-Ohmeda; Chalfont, St. Giles, United Kingdom), was performed. Animals were ventilated using pressure-controlled ventilation-volume guaranteed mode (PCV-VG) (Engström Carestation; GE healthcare, Chalfont, St Giles, Buckinghamshire, UK) with a tidal volume of 6–8 ml kg−1, a positive end-expiratory pressure (PEEP) of 5 mbar, fraction of inspired oxygen (FiO2) of 0.4, and respiratory rate adjusted to the end-tidal CO2.

Seldinger’s technique was employed for femoral vascular access after ultrasound-guided puncture to place a central venous line, a venous introducer for a pulmonary artery catheter, and an arterial introducer for a pulse contour cardiac output catheter (PiCCO; Pulsion Medical, Munich, Germany). The data from all devices were continuously monitored and stored. Balanced electrolyte fluid (BEL, Sterofundin; Braun, Melsungen, Germany) was administered at a rate of 5 ml−1 kg−1.

Sepsis induction

To induce a septic inflammatory response, a continuous infusion of LPS (E. coli Serotype O111:B4; Sigma-Aldrich, Buchs, Switzerland) was administered at a high-dose induction of 150 μg kg−1 h−1 for 1 h, followed by a maintenance dosage of 15 μg kg−1 h−1 throughout the experiment. To prevent the risk of implausible results and lung injury caused by severe hypoxemia or hypercapnia during LPS infusion, an intervention scheme was established. This scheme was based on the ARDS Network PEEP/FiO2 tables, and ventilation parameters were adjusted when the peripheral oxygen saturation dropped below 92% for 5 min.

Study protocol

Following induction and stabilisation of sepsis, the study medication was randomly assigned to one of three groups by impartial observers who were blinded to the study design. 1) Lidocaine intravenous (n = 8): 2 mg kg−1 h−1 for 1 h, followed by 1 mg kg−1 h−1

2) Lidocaine inhalative (n = 8): 2 mg kg−1 h−1 for 1 h, followed by 1 mg kg−1 h−1

3) Sham group (n = 8): NaCl 0.9%, 5 ml/h i.v. continuously.

The dosage is based on standard dosages as used for intraoperative lidocaine infusion in humans (Dunn & Durieux, 2017). The inhalation treatment group received lidocaine via a clinical nebulizer (Aeroneb ProX; Aerogen Ltd, Galway, Ireland). Blood samples were taken at baseline (BLH), 4 h (T4), and 8 h (T8) after sepsis induction. Blood gas analysis, cardiac output, and spirometry were measured at baseline and hourly after sepsis induction. Bronchial lavage was performed 8 h after sepsis induction at the lower left lobe. To maintain a mean arterial pressure above 60 mmHg, noradrenaline was administered. Glucose 40% was administered to maintain a blood glucose level above 70 mg/dL, and body temperature was measured by the PiCCO catheter, with normothermia maintained by body surface warming. The respiratory rate was adjusted to maintain an arterial pCO2 of 35–45 mmHg, and FiO2 and PEEP were adjusted to maintain SpO2 >93% according to the ARDS Network Trial.

Post mortem analysis

The experiment was ended under deep general anesthesia with an injection of 200 mg of propofol and 40 mmol of potassium chloride. The lung was extracted post-mortem in one piece and exsanguinated. The post-mortem pulmonary expressions of inflammatory markers (IL-6 and TNF-alpha) were determined in cryopreserved lung samples from the right lower lobe for mRNA analysis by real-time polymerase chain reaction (rt-PCR; Lightcycler 480 PCR System; Roche Applied Science, Penzberg, Germany). The mRNA expression was normalized to peptidylprolyl isomerase A. Lung damage was evaluated using a standardized scoring system, and the wet-to-dry ratio was determined using a predefined slice of the right upper lobe (Spieth et al., 2012; Ziebart et al., 2014).

Statistics

The data are presented as mean ± standard deviation. Statistical analyses were conducted using Sigmaplot 12.5 (Systat Software Inc, San Jose, CA, USA). The Wilcoxon test was employed to compare values before and after the intervention, whereas the two-way ANOVA on ranks followed by the Holm-Sidak method was utilized to assess intergroup differences over time. The Spearman coefficient was used to evaluate correlations. P-values less than 0.05 were considered statistically significant.

Results

In this study, 24 animals from three groups survived the observation period of 8 h after sepsis induction, while eight animals did not survive (n = 3 for sham and lidocaine i.v.; n = 2 for lidocaine p.i.).

Hemodynamics

Mean arterial pressure (MAP) and mean arterial pulmonary pressure (mPAP) were stable across all groups throughout the experiment, as shown in Table 1. Similar findings were observed for central venous pressure (CVP) and pulmonary capillary wedge pressure (PCWP) (Table 1). At baseline (BLH) and T0, heart rate (HR) and cardiac output (CO) values were comparable across all groups. However, at T4 and T8, both HR and CO values increased significantly in all groups compared to BLH, without any intergroup differences (CO: p < 0.006/p < 0.001 for T4/T8 vs. BLH for lidocaine i.v., p < 0.001 for T8 vs. BLH for lidocaine p.i., p < 0.049 for T8 vs. BLH for sham; heart rate: p < 0.001 for T4/T8 vs. BLH in all groups; Figs. 1 and 2). Extravascular lung water index (EVLWI) and global end-diastolic volume index (GEDVI) showed no significant changes throughout the experiment (Table 1).

Table 1 Hemodynamic parameters.

Parameter	Group	BLH	T0	T4	T8	
		Mean (SD)	Mean (SD)	Mean (SD)	Mean (SD)	
MAP	Lidocaine p.i.	66 (8)	69 (7)	64 (5)	62 (3)	
(mmHg)	Lidocaine i.v.	67 (8)	67 (6)	60 (4)	61 (5)	
	Sham	63 (7)	71 (8)	65 (5)	60 (7)	
mPAP	Lidocaine p.i.	8 (1)	9 (2)	14 (4)	15 (3)	
(mmHg)	Lidocaine i.v.	7 (1)	8 (1)	10 (3)	12 (2)	
	Sham	8 (1)	9 (1)	11 (2)	14 (3)	
CVP	Lidocaine p.i.	5 (1)	7 (2)	7 (2)	9 (3)	
(mmHg)	Lidocaine i.v.	6 (2)	8 (1)	9 (1)	10 (2)	
	Sham	6 (2)	9 (2)	9 (3)	10 (3)	
CO	Lidocaine p.i.	3.34 (0.4)	3.66 (0.5)	4.39 (1.1)	5.0 (1.1)*	
(l min−1)	Lidocaine i.v.	3.53 (0.4)	3.62 (0.7)	4.92 (0.9)*	6.14 (1.0)*	
	Sham	3.15 (0.5)	3.31 (0.5)	4.81 (1.1)*	5.5 (1.4)*	
PCWP	Lidocaine p.i.	8 (2)	9 (2)	7 (3)	5 (4)	
(mmHg)	Lidocaine i.v.	10 (2)	10 (2)	9 (3)	10 (4)	
	Sham	7 (2)	11 (2)	9 (2)	10 (3)	
GEDVI	Lidocaine p.i.	486 (97)	519 (116)	502 (105)	496 (96)	
(ml m−2)	Lidocaine i.v.	524 (73)	565 (138)	521 (90)	568 (136)	
	Sham	490 (75)	523 (96)	503 (105)	535 (115)	
EVLWI	Lidocaine p.i.	11 (2)	12 (2)	15 (2)	16 (3)	
(ml kg−1)	Lidocaine i.v.	11 (2)	12 (4)	15 (3)	16 (2)	
	Sham	11 (2)	12 (1)	15 (2)	15 (2)	
Notes:

* p < 0.05 vs. baseline value.

MAP, mean arterial pressure; HR, heart rate; mPAP, mean arterial pulmonary pressure; CO, caridac output; PCWP, pulmonary capillary wedge pressure; GEDVI, global endiastolic volumen index; EVLWI, entdiastolic lung water index; CVP, central venous pressure.

Figure 1 Cardiac output.

*p < 0.05 for all groups compared to BLH. BLH, baseline; Tx, Timepoints.

Figure 2 Heart rate.

*p < 0.05 for all groups compared to BLH. BLH, baseline; Tx, Timepoints.

Respiratory

The oxygen saturation (SpO2) and the fraction of inspired oxygen (FiO2) showed no differences (Table 2). At BLH and T0, the oxygen index (PaO2/FiO2) did not differ (Fig. 3). A significant drop for the PaO2/FiO2 ratio was observed at T4 and T8 in all groups (p < 0.001 for T4/T8 vs. BLH in all groups; Fig. 3). Contrary, the minute volume (MV) increased statistically over the time in all groups without any intergroup differences (p < 0.001 for T4/T8 vs. BLH in all groups; Table 2). Similar results were seen for the peak inspiratory pressure (Ppeak) and mean airway pressure (Pmean), both increased significant over time in all three groups (Ppeak: p < 0.001 for T4/T8 vs. BLH for all groups; Pmean: p < 0.024/p < 0.001 for T4/T8 vs. BLH for sham, p < 0.001 for T4/T8 vs. BLH for lidocaine i.v./p.i.; Table 2). The positive endexpiratory pressure (PEEP) showed no differences at baseline and raised statistically in all groups at T8 compared to baseline (p < 0.032 T8 vs. BLH for sham, p < 0.001 T8 vs. BLH for lidocaine p.i., p < 0.04 for T8 vs. BLH for lidocaine i.v.; Table 2). In the lidocaine intravenous group, this increase was already at T4 and remained elevated (p < 0.028 for T4 vs. BLH; Table 2). The functional residual capacity (FRC) showed only a decrease at T8 in the sham group with statistical relevance (p < 0.037 T8 vs. BLH for lidocaine i.v.; Table 2).

Table 2 Spirometry parameters.

Parameter	Group	BLH	T0	T4	T8	
		Mean (SD)	Mean (SD)	Mean (SD)	Mean (SD)	
SpO2	Lidocaine p.i.	98 (1.5)	98 (3)	96 (2)	95 (2)	
(%)	Lidocaine i.v.	98 (1)	98 (1)	97 (2)	96 (2)	
	Sham	99 (1)	98 (2)	97 (3)	95 (3)	
FiO2	Lidocaine p.i.	0.4 (0)	0.4 (0)	0.4 (0.1)	0.5 (0.1)	
(%)	Lidocaine i.v.	0.4 (0)	0.4 (0)	0.4 (0.1)	0.5 (0.1)	
	Sham	0.4 (0)	0.4 (0)	0.4 (0.1)	0.5 (0.1)	
FRC	Lidocaine p.i.	575 (190)	520 (172)	308 (191)	543 (202)	
(ml)	Lidocaine i.v.	580 (170)	570 (165)	487 (151)	423 (107)	
	Sham	574 (154)	515 (122)	440 (131)	393 (178)*	
MV	Lidocaine p.i.	6.4 (1.0)	6.6 (1.0)	7.4 (1.0)*	8.4 (1.0)*	
(l min−1)	Lidocaine i.v.	6.2 (1.0)	6.6 (0.5)	7.4 (0.5)*	8.1 (0.5)*	
	Sham	6.3 (0.5)	6.4 (0.5)	7.3 (1.0)*	7.8 (1.0)*	
Ppeak	Lidocaine p.i.	15 (2)	19 (4)	29 (7)*	30 (5)*	
(mbar)	Lidocaine i.v.	14 (1)	16 (2)	24 (6)*	26 (7)*	
	Sham	14 (2)	17 (3)	24 (4)*	28 (5)*	
Pmean	Lidocaine p.i.	8 (1)	9 (2)	14 (4)*	15 (3)*	
(mbar)	Lidocaine i.v.	8 (1)	9 (1)	13 (4)*	13 (4)*	
	Sham	8 (1)	9 (1)	11 (2)*	14 (3)*	
PEEP	Lidocaine p.i.	5 (0)	5 (0)	6 (3)	9 (2)*	
(cm H2O)	Lidocaine i.v.	5 (0)	5 (0)	7 (3)*	8 (3)*	
	Sham	5 (0)	5 (0)	5 (2)	8 (3)*	
Notes:

* p < 0.05 vs. baseline value.

SpO2, oxygen saturation; PaO2, arterial oxygen; FiO2, fraction of inspired oxygen; PaO2/FiO2, oxygen index; FRC, functional residual capacity; MV, minute volume; Ppeak, peak inspiratory pressure; Pmean, mean airway pressure; PEEP, positive end-expiratory pressure.

Figure 3 PaO2/FiO2 ratio.

*p < 0.05 for all groups compared to BLH. BLH, baseline; Tx, Timepoints.

Laboratory

Throughout the experiment, there were no significant differences in measured lactate and potassium values among all groups (Table 3). Similarly, pH values were comparable between the baseline measurement (BLH) and time point T0 for all groups (Table 3). However, at time points T4 and T8, all groups showed a significant decrease in pH compared to BLH, with no intergroup differences observed (p < 0.001 for T4/T8 vs. BLH for all groups; Table 3). The base excess (BE) showed a similar trend, with all groups exhibiting a significant decrease at T4 and T8 compared to BLH (p < 0.001 for T4/T8 vs. BLH for all groups; Table 3). The arterial oxygen pressure (PaO2) decreased over time in all three groups, with statistically significant differences observed at T4 and T8 (p < 0.001 for T4/T8 vs. BLH for all groups; Table 3). In contrast, the arterial carbon dioxide pressure (PaCO2) increased over time for all groups (p < 0.002/p < 0.001 for T4/T8 vs. BLH for lidocaine p.i.; p < 0.004/p < 0.006 for T4/T8 for sham; p < 0.04/p < 0.003 T4/T8 vs. BLH for lidocaine i.v.; Table 3). At time points T4 and T8, all groups showed a significant decrease in leucocyte count compared to the baseline measurement (BLH), with no significant intergroup differences observed (p < 0.008/p < 0.006 for T4/T8 vs. BLH for sham; p < 0.001 for T4/T8 vs. BLH for lidocaine i.v./p.i.; Table 4). Similar results were observed for thrombocyte count (Fig. 4). Hemoglobin levels remained stable over time and showed no statistically significant differences between groups (p < 0.001 for T4/T8 vs. BLH for all groups; Table 4).

Table 3 Blood gas analysis.

Parameter	Group	BLH	T0	T4	T8	
		Mean (SD)	Mean (SD)	Mean (SD)	Mean (SD)	
pH	Lidocaine p.i.	7.50 (0.02)	7.47 (0.04)	7.41 (0.07)*	7.37 (0.05)*	
	Lidocaine i.v.	7.53 (0.04)	7.48 (0.01)	7.38 (0.06)*	7.40 (0.04)*	
	Sham	7.46 (0.07)	7.47 (0.03)	7.39 (0.09)*	7.38 (0.09)*	
BE	Lidocaine p.i.	4.3 (2.1)	4.9 (1.4)	1.8 (1.9)*	1.3 (2.7)*	
(mmol l−1)	Lidocaine i.v.	5.3 (2.4)	4.6 (1.9)	0.7 (2.6)*	1.4 (1.6)*	
	Sham	3.0 (2.9)	3.8 (1.9)	−1.6 (4.3)*	1.3 (4.1)*	
paCO2	Lidocaine p.i.	35 (3)	39 (3)	44 (7)*	49 (7)*	
(mmHg)	Lidocaine i.v.	35 (2)	37 (2)	43 (6)*	44 (5)*	
	Sham	37 (4)	38 (1)	42 (6)*	46 (6)*	
PaO2	Lidocaine p.i.	204 (18)	178 (37)	93 (25)*	76 (18)*	
(mmHg)	Lidocaine i.v.	215 (21)	199 (30)	126 (30)*	79 (29)*	
	Sham	202 (18)	186 (22)	101 (24)*	81 (12)*	
Potassium	Lidocaine p.i.	3.8 (0.3)	4.1 (0.4)	4.8 (0.4)	4.9 (0.6)	
(mmol l−1)	Lidocaine i.v.	3.8 (0.2)	3.9 (0.3)	4.2 (0.3)	4.5 (0.5)	
	Sham	3.8 (0.4)	4.1 (0.3)	4.3 (0.5)	4.6 (0.6)	
Lactate	Lidocaine p.i.	1.4 (1.2)	1.6 (0.5)	2.8 (1.0)	1.5 (1.0)	
(mmol l−1)	Lidocaine i.v.	0.9 (0.5)	1.5 (0.6)	2.8 (2.3)	1.9 (5.1)	
	Sham	1.0 (0.3)	1.4 (0.6)	2.6 (0.8)	1.7 (0.8)	
Notes:

* p < 0.05 vs. baseline value.

BE, base excess; PaCO2, arterial carbon dioxide; PaO2, arterial oxygen.

Table 4 Laboratory parameters.

Parameter	Group	BLH	T4	T8	
		Mean (SD)	Mean (SD)	Mean (SD)	
	Lidocaine p.i.	17.10 (4.95)	1.59 (0.66)*	2.99 (1.65)*	
Leucocytes	Lidocaine i.v.	13.41 (3.74)	1.14 (0.19)*	2.38 (1.09)*	
(%)	Sham	12.65 (4.08)	1.23 (0.47)*	2.03 (0.47)*	
	Lidocaine p.i.	9.27 (0.53)	9.61 (0.64)	9.22 (0.71)	
Hemoglobin	Lidocaine i.v.	9.22 (0.58)	9.75 (0.58)	9.58 (0.89)	
(%)	Sham	9.63 (0.56)	10.02 (0.89)	9.30 (1.08)	
Note:

* p < 0.05 vs. baseline value.

Figure 4 Thrombocytes.

*p < 0.05 for all groups compared to BLH. BLH, baseline; Tx, Timepoints.

Post mortem analysis

The wet-to-dry ratio exhibited no significant differences between groups (6.21 ± 1.03 for p.i. vs. 6.77 ± 1.57 for i.v. vs. 5.19 ± 0.80 for sham). Further, the lung damage score confirmed the lung damage but showed no differences between the groups (Fig. 5, exemplary lung tissue from one lidocaine i.v. pig). The mRNA expression of TNF-alpha and IL-6 in lung tissue was lower in both the p.i. and i.v. groups compared to the sham group, although this difference was not statistically significant (Fig. 6).

Figure 5 Lung tissue.

Exemplary histopathological lung tissue (lidocaine i.v.). 1: alveolar oedema. 2: alveolar rupture. 3: hemorrhage.

Figure 6 Pulmonary mRNA expression of TNF-alpha and IL-6.

PPIA, Peptidylprolyl Isomerase A.

Discussion

In this study, the proposed anti-inflammatory effects of the local anesthetic lidocaine in a sepsis induced ARDS in pigs were investigated. Lidocaine was administered in two ways: intravenous and per inhalation. The sepsis induced ARDS model using LPS was chosen due to its high reproducibility and suitability in pigs (Matute-Bello, Frevert & Martin, 2008; Matute-Bello et al., 2011). The model produced common septic-like hemodynamic alterations such as an increase in heart rate and elevated cardiac output in the hyperdynamic septic state. Additionally, a significant decrease in leucocytes and thrombocytes, as required in the sepsis-related organ failure assessment (SOFA) score to screen for sepsis, was observed (Vincent et al., 1996). The present experiment also demonstrated sepsis-induced pulmonary functional impairment. The methods utilized in this study have been previously employed to investigate and demonstrate diverse aspects of sepsis, including but not limited to the evaluation of the lung-protective and anti-inflammatory properties of novel inhalation agents, the analysis of the effects of distinct ventilation protocols, and the assessment of the impact of sepsis on the endothelial glycocalyx (Ziebart et al., 2014; Thomas et al., 2022; Hartmann et al., 2014).

Intravenous administration of lidocaine has been shown to suppress the inflammatory response in a rat model of acute lung injury induced by cecal ligation and puncture (CLP) (Zhang et al., 2017). The continuous intravenous administration of lidocaine is common in human medicine and offers some advantages. Despite stable hemodynamic conditions, the continuous administration of lidocaine is associated with increased and stable blood lidocaine levels compared to the bolus injection (Kintzel, Knol & Roe, 2019). Additionally, low concentrations of lidocaine have been demonstrated to reduce anoxic damage. The therapeutic effects of lidocaine are mediated through the receptor for advanced glycation end products (RAGE) and the downregulation of the nuclear factor kappa-light-chain-enhancer of activated B cells (NF-κB) and mitogen-activated protein kinase (MAPK) signaling pathways (Zhang et al., 2017; Ma, Yan & He, 2022). These pathways have been identified as key regulators for the release of various inflammatory mediators, such as tumor necrosis factor alpha (TNF-α) and interleukin-6 (IL-6), which play a vital role in the pathogenesis of acute lung injury (Bos & Ware, 2022).

Nebulized administration of lidocaine has been observed to cause pathophysiological reactions in the lungs, such as peribronchial eosinophil and neutrophil infiltration, subepithelial fibrosis, increased collagen and mucus content, matrix metalloproteinase-9 activity, and elevated levels of interleukin-4 (IL-4), interleukin-5 (IL-5), interleukin-13 (IL-13), and eotaxin-1 (Serra et al., 2012). These effects may be attributed to the local anesthetic’s anti-inflammatory properties. Additionally, lidocaine has been suggested as a potential therapy for Coronavirus Disease-2019 (COVID-19) due to its capacity to reduce cytokine levels, protect the lungs, and lower morbidity and mortality (Ali & El-Mallakh, 2020).

Unfortunately, the study did not demonstrate any statistically significant differences in the mRNA expression of TNF-alpha and IL-6 in lung tissue. Moreover, no significant decrease in systemic inflammatory parameters, such as lactate levels, was observed.

One possible explanation is that the duration of our experiment, which was set at 8 h, may have been too short to observe significant changes. In contrast, the aforementioned study had a longer duration of 12 h, possibly allowing for a more sensitive analysis of the transcriptional regulation of inflammatory markers (Chen et al., 2018). Additionally, it should be noted that Chen et al. (2018) measured the concentrations of mediators in the bronchoalveolar lavage fluid (BALF) rather than in lung tissue, and there is a lack of reliable and comparable data on different concentrations in both compartments in the literature. Furthermore, the dosage of lidocaine used in their study was higher, up to 5 mg kg−1. Such high dosages are unusual when lidocaine is used in human medicine. Therefore, in the present study, we oriented ourselves to a dosage that has been safely and well investigated in the context of intravenous administration in the perioperative setting in humans (Dunn & Durieux, 2017). Since we could not prove the results shown by Chen et al. (2018) in our “low-dose study”, this underlines the assumption that the reported results are dose-dependent.

The present study did not observe the previously reported reduction in vascular permeability and inhibition of edema formation after intravenous and systemic administration of lidocaine (Caracas et al., 2009). In all groups, a slight increase in EVLWI was measured, possibly due to septic rupture of the alveolo-capillary unit, which contributed to the restriction of lung function, as indicated by the decreased PaO2/FiO2 ratio. This non-cardiogenic edema seen in ARDS is an independent risk factor for mortality (Jozwiak et al., 2013). Tight junctions, gap junctions, and adherens junctions are critical proteins that ensure pulmonary homeostasis and transcapillary fluid management. Inflammation and oxidative stress can target all of them, resulting in apoptosis mediated by an upregulation of NF-κB (Herold, Gabrielli & Vadász, 2013; Bhattacharya & Matthay, 2013). Lidocaine, however, failed to exhibit previously observed properties of membrane and cell stabilization at the level of the alveolo-capillary unit (e.g., elevated EVLWI) via inhibition of apoptosis by attenuating the p38 MAPK pathway (Ma, Yan & He, 2022).

Activated platelets are believed to play a crucial role in the pathogenesis of inflammation, sepsis, and sepsis-associated acute respiratory failure (Kiyonari et al., 2000). Platelet-leukocyte aggregation (PLA), the interaction with leukocytes, has been reported as a potential marker for sepsis and thromboembolism in critically ill patients. Reports suggest that local anesthetics, especially lidocaine, modulate platelet activation and aggregation. In vitro and in vivo studies have shown that lidocaine reduces inflammatory injury caused by reperfusion, endotoxin-, and hypoxia-induced injury at an early stage, with a stabilization of platelet counts (Huang et al., 2009; Lo et al., 2001; Swanton & Shorten, 2003). One possible mechanism is the inhibition of the ADP-induced P-selection expression for PLA (Huang et al., 2009). In this study, a drop in platelet count associated with sepsis-like changes was observed. Unfortunately, the reported effects of stabilizing platelets and leukocytes were not observed. But both studies are difficult to compare because the dosages used were different. The dosages used by Huang et al. (2009) were clinically relevant for local application but not for intravenous application. The dosage of lidocaine and the observed anti-inflammatory effects appear to be particularly important in all reports.

Several limitations to this study should be considered: (1) 8 h of experimentally induced ARDS only reflect the earliest phase of pathophysiological changes in ARDS. These circumstances are due to local regulations. (2) To reduce confounders in analyzing the results, only one gender was used, which is a non-clinical and unreal scenario. (3) The serum levels of inflammatory markers should have been determined for better comparability of the study results. Further, the serum concentrations of lidocaine, especially in the inhalative group, should have been measured. (4) The statements about the effect on the thrombocytic level can only be used indirectly and to a limited extent. A differentiated thrombocyte examination was not carried out.

Conclusion

Unfortunately, the present study did not provide evidence to support the previously reported anti-inflammatory effects of lidocaine in a porcine model of septic ARDS, irrespective of its route of administration (inhalation and intravenous). Future investigations should focus on extending the duration of the study, conducting more detailed anti-inflammatory assessments, and examining different dosages of lidocaine. The potential role of lidocaine as a therapeutic agent for acute lung injury patients remains uncertain.

Supplemental Information

Supplemental Information 1 Lidocaine in ARDS Dataset.

Click here for additional data file.

The authors thank Mrs. Dagmar Dirvonskis and Mr. Bastian Duenges for support in logistics and laboratory organization. The manuscript was revised and proofread by ChatGPT Feb13 (OpenAI; USA).

Additional Information and Declarations

Competing Interests

Author Contributions

Animal Ethics

Data Availability

Erik K. Hartmann is an Academic Editor for PeerJ.

René Rissel analyzed the data, prepared figures and/or tables, authored and approved the final draft.

Christian Moellmann conceived and designed the experiments, performed the experiments, authored or reviewed drafts of the article, and approved the final draft.

Victoria Albertsmeier performed the experiments, prepared figures and/or tables, and approved the final draft.

Miriam Renz analyzed the data, authored or reviewed drafts of the article, and approved the final draft.

Robert Ruemmler analyzed the data, authored or reviewed drafts of the article, and approved the final draft.

Jens Kamuf performed the experiments, authored or reviewed drafts of the article, and approved the final draft.

Erik K. Hartmann conceived and designed the experiments, performed the experiments, authored or reviewed drafts of the article, and approved the final draft.

Alexander Ziebart conceived and designed the experiments, performed the experiments, analyzed the data, prepared figures and/or tables, authored or reviewed drafts of the article, and approved the final draft.

The following information was supplied relating to ethical approvals (i.e., approving body and any reference numbers):

The State and Institutional Animal Care Committee (Rhineland-Palatinate, Koblenz, Germany) approved the study (ID G16-1-015).

The following information was supplied regarding data availability:

Raw data are available in a Supplemental File.

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
