# Peer review of "Clinical dosage of lidocaine does not impact the biomedical outcome of sepsis-induced acute respiratory distress syndrome in a porcine model"

_PeerJ, doi:10.7717/peerj.15875_

## Round 0.1 · original submission · Major Revisions

Dear Authors,

Thank you for submitting your critical work to us. We have read it carefully and received the review reports from our peers. We appreciate the effort you put into the study, and we would like to provide you with some feedback on improving it further.

Two reviewers appreciated your work, while two others recommended that the study be further improvised on both experiment and literature. There are a few major points that we would like to highlight:

Reviewer 1 suggested adding histopathology, even though it is a negative report. Reviewer 2 commented on the structure of the manuscript, specifically pointing out that the diagrams and results section needs improvement. They also questioned the contradiction from previous reports. Reviewer 3 also raised a flag about the study contradicting previous reports. Reviewer 4 questioned the in vivo part, which is the major finding of the study.

Based on the peer-review reports, we have decided to go for a MAJOR revision rather than rejection. We urge you to address the issues point-by-point, as outlined in the report. This way, we can work together to improve your study and, hopefully, get it published soon after the peer-reviewers acceptance.

I appreciate your cooperation.

Kindly,
Dr. Nagendran Tharmaingam
Handling Editor.

Reviewer 1 ·

Basic reporting

The manuscript is generally clear and easy to understand. The article structure needs to be improved. The abstract routinely included subtitles of background/objective, methods, results and conclusions, which provides a clear, concise overview of the paper's main points and should be written in a way that is self-contained. For the methods section, measurements are suggested to be categorized for a clearer description. For results section, subtitles are suggested for a clearer description.

Experimental design

This study evaluated the effect of intravenous and inhaled lidocaine in an animal model of sepsis induced lung injury. Their findings showed there were no significant improvement of lung injury after lidocaine treatment, no matter intravenously used or inhaled. Though it’s a study with negative findings, it still provided some valuable scientific significance. There are some comments for a deeper understanding and improvement of the study.

1. How the dose of lidocaine through intravenous or inhalation (2mg/kg/h) was determined? Please provide supporting reference.

2. How was the sample size of the study determined? Except animals sacrificed during the 8hr observation time, how many animals in each group were left for the comparison? How was the survival difference among the three groups?

3. As stated in the limitation, the observation time of the study is relatively short.

4. As stated in the Methods section, “lung damage was evaluated using a standardized scoring system, and wet-t-dry ratio was determined using a predefined slice of the right upper lobe.” Please provide corresponding results.

Validity of the findings

Please provide histopathological images of sepsis induced acute lung injury in each group and corresponding representative images after lidocaine intervention.

·

Basic reporting

- The data are reported as numbers (Tables 1-3) instead of figures. Plots are much easier to perceive as compared to raw data tables, hence tables 1-3 should be converted to figures.
- Bar charts (Fig. 1) are an outdated form of graphical data presentation. Beeswarm plots or box plots should be used instead which represent the spread, outliers, and individuality in the data in a much more effective manner.
- The manuscript is extremely short and lacking many important details regarding the background- especially previous works and the authors' rationale for their currently reported experimental design in the context of previous work.
- The methods are appropriately reported, except for the rationale as per above point.
- The results is just one long paragraph, which is basically the tabular data written in text form.
- The discussion answers some questions raised throughout the manuscript, however, it reveals that the authors' experimental design is very different (where it matters) from the previous works.

Experimental design

The authors discuss that they failed to observe any of the Lidocaine-mediated effects which were reported by previous works. They attribute the absence of these effects to a lower dosage of lidocaine and time-points of their experimental observation. It is unclear why the authors chose dosage and time-points different from the previous works. This erroneous experimental design yielded insignificant results, even across conditions where the effect of lidocaine is reported to be significant.

Validity of the findings

Since the authors failed to reproduce the effects reported by multiple earlier works, the findings are not very useful. e.g., the comparisons are valid at the lidocaine dosage used by the authors in their experiments, but this dosage does not make any effect on the outcome (unlike previous works which used a higher dosage) and hence is not biomedically relevant.

Additional comments

The title is misleading, it should report the main result. Suggested title: "A low dosage of lidocaine does not impact the biomedical outcome of sepsis-induced acute respiratory distress syndrome".

Reviewer 3 ·

Basic reporting

In this manuscript by Rissel and colleagues, the authors attempt to study the effect of lidocaine delivered via inhalation in an LPS-induced sepsis porcine model. The manuscript is well-written and the language is succinct and clear.

I have a few minor suggestions to improve the manuscript:
1. While the results have been described in detail, the data representation could be better. The manuscript includes data in the form of multiple tables, but only one experimental data has been plotted into a graph. I recommend converting all the tabular data that have any significant differences to report into graphs. This would make the data easier to interpret.
2. Figure 1 requires a caption followed by description.

Experimental design

The introduction and methods section are well-described, clearly stating the aim and need for this study.

However, as shown in the results and as pointed out by the authors, the findings of this manuscript are inconclusive. Because Chen et al. and Huang et al., have shown intravenous lidocaine to have positive effects, I would recommend the authors to consider matching or increasing the duration, dosage and use the same analyzed tissues to compare the different modes of lidocaine delivery (per inhalationem vs intravenous). This will hopefully replicate the intravenous lidocaine effects and give insight into the potential possibility of benefits from delivery by inhalation.

Validity of the findings

While the interpretation of the experiments performed and data interpreted is valid and fair, there are deficits in the experiment design that make it difficult to draw conclusive results pertaining to the primary question asked in this study.

Reviewer 4 ·

Basic reporting

The manuscript by Rissel et al is proposed to address the effect of a common local anesthetic, Lidocaine in sepsis-induced acute lung injury using pig models. The authors use both intravenous and inhalation modes of administration and demonstrate that inhalation method has the same effects as intravenous method. However, the authors could not reproduce previous findings by other groups showing that Lidocaine has anti-inflammatory and lung protective effects. Although the study attempts to identify the effect of Lidocaine using LPS-induced pig model, there are several limitations that make it not suitable for publication.

Experimental design

The comments are,
1. The experimental design has not satisfactorily explained. The authors do not mention how long LPS has been administered to induce sepsis and get the ARDS phenotype. There is no confirmation through histology of lung that there is ARDS phenotype.
2. The method section states that the study used a total of 32 pigs and results mention that out of 32 animals, 8 animals had not survived after the procedure. How did you distribute animals across the three groups? There is no clear results on how many animals per group has been assigned either in methods or results section. When authors mention ‘distributed equally over all three groups’ (line 167), it is unclear whether they talk about survived animals or not survived ones.
3. T4 and T8 are assumed to be time points. It has not been expanded.
4. The hemodynamic and pulmonary measurements alone do not substantiate the results. There is no mechanistic insights to claim that both intravenous and inhaled effects are same.
5. The author’ claim on the limitation of the approach including the duration of treatment and concentration of Lidocaine is something needs to be considered for the improvement of this study.
6. In the introduction, there is no reference given for lines 84-87. Also, line 87 contradicts the fact that Lidocaine is anti-arrhythmic. The reference given is a general review for that line not specific.
7. Proofreading using ChatGPT is a good option. However, depending on that alone may not be sufficient.

Validity of the findings

Nothing to add here

---

## Round 0.2 · Minor Revisions

Dear Authors,

Thank you for your revised work submitted to us. We are glad reviewers 1 and 2 are happy to receive your responses. However, I believe reviewer 3 raised a minor concern that you can easily address, I believe firmly. Please let us know if you have any questions and we are looking forward to seeing your responses to reviewer 3.

Kindly,
Dr. Nagendran Tharmalingam
Handling Editor.

Reviewer 1 ·

Basic reporting

The authors have addressed the comments satisfactorily.

Experimental design

The authors have addressed the comments satisfactorily.

Validity of the findings

The authors have addressed the comments satisfactorily.

·

Basic reporting

The authors' responses to the reviewer comments as well as the revisions in the manuscript clarifies the authors' motivations, limitations, and the resultant experimental design. Specifically, the authors' choice of the lidocaine dosage now makes much more sense, in light of the clinical relevance that is now added to the manuscript. The added discussion of their results in contrast with earlier work also clarifies the results. Overall, the new version of the manuscript is impactful and important for clinical decisions.

I appreciate the figures summarizing the tables.

Experimental design

commented above

Validity of the findings

commented above

Additional comments

I suggest a slight tweak to my earlier suggestion of the title. Now that the authors have defended their experimental design and dosage of lidocaine, I believe replacing "low dosage" in the title with "Clinical dosage" would make the title more impactful.

Reviewer 3 ·

Basic reporting

I am happy to see improved representation of the data by the addition of figures. The language and structure of the manuscript is also much improved.

Experimental design

Deficits in the experimental design remain, as the authors only provide an explanation as to why they chose the low-dose for their experiment. No comments have been made about the duration of dose, which I think should be matched with previous reports.

Validity of the findings

The objective of this study, as stated by the authors is to- "investigate whether the inhalation of lidocaine can produce effects comparable to those achieved with intravenous administration in a pig model of lipopolysaccharide (LPS)-induced sepsis and acute lung injury."
The conclusion stated is "the present study did not provide evidence to support the previously reported anti-inflammatory effects of lidocaine in a porcine model of septic ARDS, irrespective of its route of administration (inhalation and intravenous)."
The two statements clearly show that the manuscript is unable to address the primary objective/question because even the comparative conditions didn't work.
The inconclusive results generated from the experiments shown here do not allow us to draw any conclusions, so unfortunately, I do not see a need for this work to be published in its current state.

---

## Round 0.3 · accepted · Accept

Dear Authors,

Thank you for your patience through the rigorous review process and patience. We are happy to receive your revision and glad to inform you that I accept your work to be published in PeerJ after I agreed on the responses between reviewers and authors. The production house will contact you if they need any further assistance.

Best wishes.
Dr. Nagendran Tharmalingam,
Handling Editor.